# Bioadhesion on Textured Interfaces in the Human Oral Cavity—An In Situ Study

**DOI:** 10.3390/ijms23031157

**Published:** 2022-01-21

**Authors:** Ralf Helbig, Matthias Hannig, Sabine Basche, Janis Ortgies, Sebastian Killge, Christian Hannig, Torsten Sterzenbach

**Affiliations:** 1Max Bergmann Center of Biomaterials, Leibniz-Institut für Polymerforschung, Hohe Straße 6, 01069 Dresden, Germany; helbig@ipfdd.de; 2Clinic of Operative Dentistry, Periodontology and Preventive Dentistry, University Hospital, Saarland University, 66421 Homburg, Germany; Matthias.Hannig@uks.eu (M.H.); Janis.Ortgies@uks.eu (J.O.); 3Clinic of Operative and Pediatric Density, Medical Faculty Carl Gustav Carus, Technische Universität Dresden, Fetscherstraße 74, 01307 Dresden, Germany; Sabine.Basche@ukdd.de (S.B.); Christian.Hannig@ukdd.de (C.H.); 4Institute of Semiconductor and Microsystems, Chair of Nanoelectronics, Technische Universität Dresden, 01609 Dresden, Germany; Sebastian.Killge@tu-dresden.de

**Keywords:** textured surfaces, bioadhesion, oral, pellicle, microbiota, restorative dentistry

## Abstract

Extensive biofilm formation on materials used in restorative dentistry is a common reason for their failure and the development of oral diseases like peri-implantitis or secondary caries. Therefore, novel materials and strategies that result in reduced biofouling capacities are urgently sought. Previous research suggests that surface structures in the range of bacterial cell sizes seem to be a promising approach to modulate bacterial adhesion and biofilm formation. Here we investigated bioadhesion within the oral cavity on a low surface energy material (perfluorpolyether) with different texture types (line-, hole-, pillar-like), feature sizes in a range from 0.7–4.5 µm and graded distances (0.7–130.5 µm). As a model system, the materials were fixed on splints and exposed to the oral cavity. We analyzed the enzymatic activity of amylase and lysozyme, pellicle formation, and bacterial colonization after 8 h intraoral exposure. In opposite to in vitro experiments, these in situ experiments revealed no clear signs of altered bacterial surface colonization regarding structure dimensions and texture types compared to unstructured substrates or natural enamel. In part, there seemed to be a decreasing trend of adherent cells with increasing periodicities and structure sizes, but this pattern was weak and irregular. Pellicle formation took place on all substrates in an unaltered manner. However, pellicle formation was most pronounced within recessed areas thereby partially masking the three-dimensional character of the surfaces. As the natural pellicle layer is obviously the most dominant prerequisite for bacterial adhesion, colonization in the oral environment cannot be easily controlled by structural means.

## 1. Introduction

Oral diseases (e.g., caries, gingivitis, periodontitis, and root canal infections) belong to the most common causes of medical treatment affecting most people during their whole lifespan [1]. In general, they are directly or indirectly caused by biofilm formation of members of the oral microbiota on both shedding (e.g., mucosa) or non-shedding (e.g., teeth) surfaces [2]. Extensive biofilm formation can also occur on restorative dental materials, resulting in various conditions, such as secondary caries, mucositis, peri-implantitis, or ultimate failure of the medical devices [3]. Hence, there is a great interest in the development of novel materials and strategies to reduce biofilm formation on restorative dental materials [3].

Many approaches outside of the oral cavity focus on regularly structured or rough surfaces, though results are not always conclusive [4]. Generally, structures larger than the size of a bacterium rather promote bacterial adherence due to an increased surface area and protection of the microorganisms from shear forces and mechanical stress [3,5]. Dimensions close to the cells’ size result in substrate feature distances that fit well to the cell shape; the available area for adhesion is large, and bacterial colonization is strongly supported [6]. Some structures similar to those used in the present study reduced bacterial adhesion within a micrometer scale with preferential adherence in recessed surface areas [7,8]. Dimensions smaller than the size of a typical bacterium generally reduce microbial colonization. In vitro assays with holes, post, and line structures slightly below the size of a typical bacterium (<1 µm) reduced retention of both Gram-positive and Gram-negative bacterial species. In contrast, structure dimensions larger than a typical bacterium (>1 µm) lead to unchanged or increased retention compared to unstructured surfaces [6]. Additionally, a submicrometer-sized roughness smaller than the size of a bacterium (0.2–1 µm) delayed biofilm formation [9]. Submicrometer-sized roughness generated by an irregular three-dimensional layer of silicone nanofilaments suppressed adhesion of *E. coli* on glass slides [9]. Even for the fungal pathogen, *Candida albicans*, a reduced biofilm formation is reported on coatings generated by spherical silica particles in the sub-cell size range of 0.5–2 µm compared to larger dimensions of 4–8 µm, which were in the range of the mean size of the cells in their yeast cell form [10]. Furthermore, sub-cell-sized structures generally tend to have inhibiting effects, whereby the actual length scale seems to depend on the microbial strain.

However, promising strategies often fail in the oral environment [3]. A major caveat and challenge for the development of anti-adhesive surfaces is the formation of the oral acquired pellicle [11,12,13,14]. This is a thin layer of proteins and other macromolecules composed of salivary as well as pro- and eukaryotic cellular components [15]. Both mucosal surfaces and teeth, as well as restorative materials accessible to saliva, are almost instantly covered by the pellicle [12]. It has both anti-microbial as well as biofilm formation-promoting properties. Importantly, the pellicle can level out surface roughness or mask physico-chemical characteristics (e.g., surface free energy) [16,17,18,19]. For example, an in vitro-formed pellicle by incubation with saliva eliminated the effect of surface roughness on biofilm formation by several *Streptococcus spp.* [20]. Other studies using in vitro-formed pellicles to mimic the oral environment also found that pellicle formation has complex effects on initial adhesion and biofilm formation of three strains implicated in infectious processes on various rough substrates [21]. While biofilm formation was reduced in all three strains after conditioning of the substrates in saliva, compared to unconditioned specimens, the initial adhesion was in part increased depending on the strain. Biomimetic studies of the anti-adhesive submicro-scaled cuticle of springtails showed promising biofouling resistance under in vitro conditions, whereas these characteristics were mostly lost in the oral cavity within a few hours [22]. However, there were some findings of a significant correlation between surface roughness and initial surface colonization (120 min) also under in situ conditions [23], but not for longer exposure times (24 h) [24].

Systematic studies using defined and regular surface structures with varying dimensions over a larger range are necessary to finally elucidate the impact of topographical length scales on microbial colonization within the oral cavity. Therefore, the purpose of the present work was to gain a better understanding of the interplay between surface structures, the acquired pellicle, and the oral microbiota and clarify whether biofilm formation can be significantly controlled by this means at all using a well-established in situ model [15,22,25,26,27].

## 2. Results

### 2.1. Pellicle Formation on PFPE

As a basic material for all textured surfaces, the hydrophobic polymer perfluorpolyether (PFPE) was utilized, as it is an excellent chemically inert and non-toxic molding material replicating the smallest surface features of a master structure in high detail. In the first step, it was tested whether a functional pellicle was formed on this surface. Therefore, specimens fabricated from PFPE were fixed on splints and worn for 30 min bucally in the oral cavity. As a control, specimens made from enamel were exposed to the oral cavity in parallel fixed to the same splints. Afterward, activities of amylase and lysozyme, two key enzymes in saliva, were measured (Figure 1a,b). The enzymes can be immobilized rather tightly or loosely within the oral acquired pellicle. Hence, activities within an immobilized and desorbed fraction after removing the specimens from the substrate were compared. Enzymatic activities of both amylase and lysozyme in the immobilized fraction did not differ significantly from pellicles formed on enamel or PFPE. In addition, enzyme activities within the desorbed fraction were also comparable in both substrates, suggesting that the enzymes are similarly tightly fixed within the pellicle. These results suggest that a natural pellicle was formed on the synthetic polymer PFPE, which could be confirmed by TEM imaging (Figure 1c). Here, an electron-dense basal layer could be visualized covered by the more complex and thicker (approximately 700 nm) globular layer. Both the ultrastructure and thickness of the pellicle are comparable to published studies on the ultrastructure of the pellicle formed on enamel [11,28,29].

### 2.2. Larger Textured Surfaces Do Not Influence Biofilm Formation

In a first approach, PFPE-specimens with structures of different geometry (lines, holes, and pillar structures) were fabricated. The dimensions of the different structures were 2 µm and 5 µm, thereby slightly larger than the size of typical bacteria, each with a depth of 2 µm. The substrates were manufactured using silicon master structures (Appendix A).

The different specimens, as well as plain PFPE controls, were bucally exposed to the oral cavity for 8 h with three subjects and three independent repeats. Then the number of adherent bacteria on the specimens was evaluated by DAPI-staining and microscopical evaluation. Since colonization can vary strongly between repeats and subjects, we decided to depict the relative number of adherent bacteria normalized to the average bacterial numbers on plain PFPE rather than absolute values. This normalization procedure reduced the influence of variation between subjects and day-to-day variation within repeats with the same subjects.

Here, only the 5 µm line structures had significantly higher numbers of adherent cells compared to the control (plain PFPE), while all other micro-structured specimens (lines, holes, or pillars with dimensions of 2 or 5 µm) showed no statistically significant difference to the control (Figure 2). However, there was a tendency that colonization of the surfaces with 2 µm dimensions—and even more pronounced with 5 µm dimensions—was generally slightly higher on the micro-structured surfaces compared to the control. Bacterial colonization was predominantly observed within the structural cavities for all feature types, probably due to protection from hydrodynamic shear forces. Within 5 µm structures, bacteria also tended to be located at the edges of the impressions.

This suggested that micro-structured surfaces constructed from PFPE with dimensions larger than a typical bacterium did not reduce, but rather tended to result in an increased bacterial adherence.

### 2.3. Evaluation of Bacterial Colonization with Gradient Structures

Since generally no significant influence of larger feature sizes (2 µm and 5 µm) was detected, a new strategy was used. Here specimens with gradient structures of different dimensions encompassing a range smaller than bacteria, in a size range of typical bacteria and much larger sizes (0.7–29 µm feature size, 0.7–130 µm feature distance) were developed. Again, three different morphologies were used (line, hole, and pillar structures). Within each gradient morphology, the sizes of structural features were varied in one direction and distances between features in the perpendiculardirection (Appendix A). To evaluate the quality of the specimens, cross-sectional views were recorded via TEM. These showed a high regularity of the structures (Appendix A).

Again, the specimens were carried bucally for 8 h by three subjects with three independent repeats. Microbial colonization was evaluated by DAPI-staining and microscopical counting of adherent cells. Due to the large number of sub-fields, only selected sizes were evaluated, as indicated in Appendix A.

Bacterial colonization of the specimens showed large variability. Additionally, for the most part, biofilms were only one cell layer of thickness. In Figure 3, Figure 4 and Figure 5, only selected dimensions are shown representatively, while in Appendix A, all analyzed dimensions are depicted. Colonization levels were normalized to the colonization on the plain control. However, despite large variations, we could not detect significant trends or corellations between the different structure sizes. Two different ways were used to arrange the results—either groupings of equal structure distance with increasing size of structural features (Appendix A) or simply by increasing periodicity overall gradient sub-fields (Appendix A). In part, there seemed to be a decreasing trend of adherent cells with increasing periodicities and structure sizes, but this possible pattern was weak, irregular, and contained too many outliers. As for specimens with defined single dimensions (2 µm or 5 µm), bacteria were located for the most part within impressions of all three feature types. Finally, we compared all complete datasets of every structure type in one plot, i.e., all data from the line, hole, and pillar gradient arrays were taken together, respectively (Appendix A), in order to see if there was a general mean difference between these different morphologies, but no significant differences emerged.

### 2.4. Transmission Electron Microscopy of Specimens

In the next step, TEM images of the gradient specimens were prepared to visualize the formation of the oral acquired pellicle. Therefore, cross-sectional views were cut from the center of the specimens. It was observed that in all cases, a mature pellicle was formed (Figure 6); however, while a basal pellicle was formed rather uniformly, a more globular layer was pronounced within recessed areas of the structures, hence within lines, holes, or between pillars. This is probably due to hydrodynamic and mechanical forces during exposure. Hence, the three-dimensional structures were at least partially equalized and coated by the pellicle.

## 3. Discussion

In this study, the bacterial colonization of micro-structured surfaces within the oral cavity was systematically analyzed. Therefore, microstructures composed of PFPE with different dimensions and geometries were used. Previous in vitro studies suggested that regular structures or irregular surface roughnesses with dimensions larger than and similar to the size of a bacterium rather promote bacterial adherence due to protection and a highly accessible surface for adhesion, respectively, while dimensions smaller than the size of a typical bacterium inhibit or delay colonization at least in the initial phase [6,7,8,9,10,30]. However, these studies addressed general questions concerning microbial adherence or biofilm formation to different engineered surfaces in ex vivo environments without the presence of a multitude of biomolecules, such as peptides and proteins [3], or the coexistence of many different bacterial strains. In contrast, the oral cavity is one of the most complex and challenging environments within the human body, with an intricate interplay between soft and hard oral surfaces (mucosa and teeth), saliva with thousands of different biomolecules, and, finally, the oral microbiome, with hundreds of different bacterial species [2,3,5,14,31,32].

In particular, all surfaces accessible to saliva are covered almost instantaneously by the acquired pellicle. The ultrastructure of the pellicle formed on artificial restorative materials is similar to the pellicle formed on enamel [28,29]. It has been shown that the activities of enzymes within this layer did not differ significantly on these substrates, suggesting the presence of a functional pellicle [33,34]. Accordingly, we found that both the ultrastructure of the acquired pellicle and the activities of two key enzymes, amylase and lysozyme on PFPE, were also comparable with enamel [29,35]. The presence of the oral acquired pellicle can strongly alter pristine surface properties, i.e., the physico-chemical identity, and translate it after establishment into a common biological identity [16,17,18,19]. The pellicle may also potentially mask small surface topographies, which further affect the adhesion or repulsion of bacterial cells (both promoting and anti-microbial properties) and, therefore, biofilm formation [20,22].

Across all different surface structures with dimensions smaller than, in the size range of, or larger than typical bacteria compared to flat surfaces and with each other, we could not find major significant trends or patterns in the extent of bacterial colonization. The covering of the surface by the formation of a pellicle apparently reduced topographical effects found in other more simplified environments. This hypothesis is consistent with TEM observations. While a basal pellicle could be detected throughout the specimens, a more globular pellicle layer has been formed in the structural cavities, partially smoothing out structural features. This is probably caused by both mechanical and hydrodynamic processes on the exposed topmost part of the substrates. However, TEM also showed that the surface topography is not completely hidden by the pellicle layer, which is in line with the observation that the bacteria were still concentrated on the bottom edges within the structures. [36,37]. The altered globular composition of the pellicle in these areas could promote enhanced bacterial adherence [38]. This would imply that larger structures exhibit a smaller amount of adherent bacteria, because of a smaller density of edges. But a weak trend of decreasing adherence with an increasing distance of structural features could be observed only in part and was not significant due to large inter- and intra-individual variability. This trend of bacterial surface colonization might also be explained by decreasing mechanical protection with decreasing aspect ratio, however, the effect is obviously negligible.

## 4. Materials and Methods

### 4.1. Specimen Preparation

The fabrication of the silicon master structures was performed via deep reactive ion etching. The basic pattern was given by a photoresist (AZ 5214e, Microchemicals GmbH, Ulm, Germany), which was exposed to UV light through a photomask (Photronics MZD, Dresden, Germany). After the development (AZ 726 MIF) of the resist, the silicon was etched with SF_6_ during continuous passivation of the emerging structure walls with C_4_F_8_ until a depth of 2 µm was reached.

The silicone masters were molted with PDMS (Sylgard 184, Mavom, Stuttgart, Germany) within a supporting polymer frame (Perfactory Resin R11, EnvisionTEC, Gladbeck, Germany), which was fabricated via stereolithography (Asiga Max). The resulting PDMS stamp, containing the negative structure, was filled with the liquid prepolymer PFPE (Solexis Fluorolink MD700, ACOTA, Oswestry, UK) and covered with a glass slide. The fluorpolymer was UV cured under oxygen exclusion, by continuous rinsing with inert gas for at least 30 min, and finally carefully peeled out from the stamp.

### 4.2. In Situ Experiments

In situ experiments were carried out by a well-established system (Appendix A) [22,25,39,40]. For all participants, upper jaw splints were constructed from methacrylate. The specimens were first incubated for 7 days in ddH_2_O to release potential residues. Then the specimens were cleaned in 70% ethanol and autoclaved for sterilization. Afterward, the specimens were fixed at the buccal sites of the splints in the regions of the upper premolars and molars using polyvinyl siloxane impression material (President Light Body, Coltène, Altstätten, Switzerland). The splints containing the specimens were worn for 30 min (enzymatic assays) or 8 h overnight (fluorescence microscopy and TEM). The subjects (three different subjects and three independent repeats) were instructed to refrain from eating and drinking during the exposure time. For 8 h overnight incubations (fluorescence microscopy and TEM), the splints were wrapped into moist paper towels the next morning and immediately brought to the laboratory for further processing. Splints carried for enzymatic assays were processed immediately after wearing the splints for 30 min.

### 4.3. Enzymatic Assays

For enzymatic assays (amylase or lysozyme), the specimens (PFPE or enamel) were exposed intraorally for 30 min. Afterward, the specimens were rinsed with ddH_2_O to remove loosely attached biological substances or residual saliva and were then used as samples for the enzymatic assays. An immobilized activity (incubation of the specimens within the respective test solution) and a desorbed activity (removal of the specimens from the test solution and continuous measurement of desorbed enzymes from the pellicle) were determined for both amylase and lysozyme.

Amylase activity was determined as previously described [41]. Amylase hydrolyses GalG2CNP to aglycone 2-chloro-4-nitrophenolate (CNP), which can be measured photometrically at λ = 405 nm. For the assay, the specimens were incubated in amylase test solution (5 mM Galactopyranosylmaltotriosid (GalG_2_CNP), 5 mM CaCl_2_, 50 mM NaSCN, 0.03% albumin and 0.03% NaN_3_ in 50 mM MES buffer, pH 6.0) and incubated for 10 min (immobilized activity). Absorption at 405 nm was measured before (base value) and immediately after removal of the specimens from the amylase test solution with a Tecan infinite M2000 reader (Tecan, Crailsheim, Germany), and activity was determined as amylase activity in mU/cm^2^ using CNP as a standard. For the desorbed activity, the amylase test solution was incubated for an additional 10 min after removal of the specimens. Afterward, absorption of the solution was remeasured at 405 nm to determine the desorbed activity.

Lysozyme activity was also measured, as previously described [42]. Here, lysozyme activity is measured using fluorescein-labeled *Micrococcus lysodicticus* (EnzCheck lysozyme assay kit, Molecular Probes, Leiden, The Netherlands). Lysozyme activity releases fluorescein, leading to an increase in fluorescence. For the assay, the specimens were incubated with the substrate diluted in reaction buffer (0.1 M sodium phosphate, 0.1 M sodium chloride, and 2 mM sodium azide, pH 7.5) and incubated for 10 min for the immobilized activity. Fluorescence was measured using a Tecan infinite M200 reader at an excitation of γ = 485 nm and an emission of γ = 535 nm before the addition of the specimens (base value) and then in 1 min intervals for 10 min after the addition of the specimens to the test solution. Lysozyme activity was then calculated in U/cm^2^ using purified lysozyme as standards. For the desorbed activity, the assay solution was incubated for an additional 10 min after removal of the specimens. Here, fluorescence was again measured in 1 min intervals to determine the desorbed activity.

### 4.4. Fluorescence Microscopic Assays

For the fluorescent microscopical counting of microorganisms, the DNA of the microorganisms was stained with DAPI (4′,6-diamidino-2-phenylindole). In short, the specimens were rinsed with 0.9% NaCl and then covered with DAPI (working solution 1 μg/mL) and incubated for 15 min in the dark. Afterward, the DAPI solution was removed, and the specimens were washed with 0.9% NaCl and air-dried. Epifluorescence microscopic analyzes were then performed at 1000-fold magnification (Axioskop II; Zeiss). Microbes were counted manually per square from 10 randomized microscopical ocular grid fields per sample and the number of adherent cells was then calculated per area [22,43,44].

### 4.5. TEM

After 8 h of oral exposure, gradient specimens were processed for TEM analysis. After fixing in 1% glutaraldehyde/1% paraformaldehyde for 2 h, specimens were post-fixed in 2% osmium tetroxide. Dehydration took place in an ascending series of ethanol. Specimens were embedded in Araldite CY212 (Agrar Scientific, Stansted, UK) and ultrathin sections were cut using a diamond knife. The ultrathin sections were contrasted with uranyl acetate and lead citrate and analyzed in a TEM TECNAI 12 Biotwin (FEI, Eindhoven, The Netherlands). Representative micrographs were taken at magnifications of 1000-fold up to 90,000 fold.

### 4.6. Statistical Analyzes

Statistical analyzes were conducted with GraphPad. One-way ANOVA was used as a statistical tool for multiple comparisons with Tukey posthoc analyzes. *p*-values below 0.05 were considered significant.

## 5. Conclusions

In summary, we could not detect major differences between three-dimensional surfaces with different geometries and structure sizes. The complexity of the oral environment and the ability of the pellicle to mask and disguise surface properties limits the potential of common antifouling strategies. More sophisticated approaches will be necessary to accomplish the control of bioadhesion on improved dental restorative materials. If the pellicle layer is a strong promoter for biofilm formation, and its establishment cannot be prevented (except for known entropically repulsive, but non-applicable PEG/PEO-like systems [45,46]), one key to influence bacterial colonization might be to alter pellicle cohesion and/or consistency. Possible strategies could include the defined use of surface charges, such as spatially alternating polarity, zwitterionic compounds, or dynamic amphiphilic molecules [47,48,49], as most proteins sensitively interact with surface charges and, in response, change their alignment and conformation. This alternation of the first adsorbed layer might reduce binding forces to further adsorbing biomolecules, leading to a weak cohesion and, therefore, to an easy detachment of settling bacteria under a low mechanical impact, such as micro-hydrodynamics [50,51]. The fabrication of respective substrates is much more elaborated compared to the manufacturing of structured surfaces, but bioadhesion in the oral cavity obviously cannot be influenced by simple means.

## Figures and Tables

**Figure 1 ijms-23-01157-f001:**
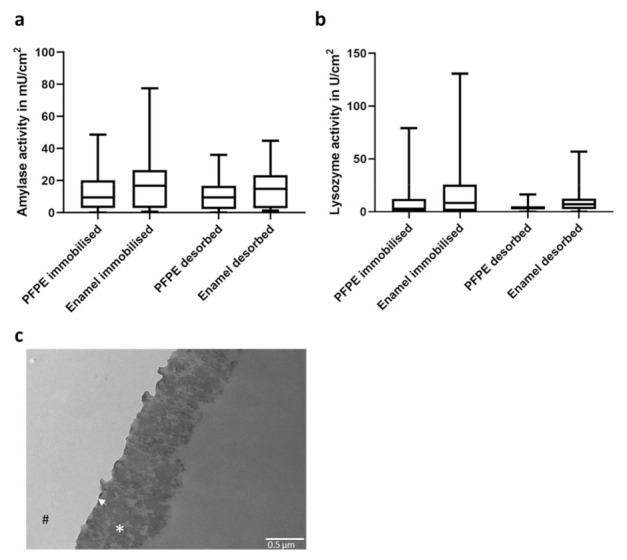
Activity of enzymes in the oral acquired pellicle. The activity of amylase (**a**) or lysozyme (**b**) was measured within the pellicle formed after 30 min in situ on PFPE or enamel. (**c**) Representative TEM-image of the oral acquired pellicle on PFPE after in situ exposure within the oral cavity for 8 h. An electron-dense basal pellicle layer (arrow) and the more complex thicker globular layer (*) can be distinguished. # PFPE side; magnification 30,000-fold.

**Figure 2 ijms-23-01157-f002:**
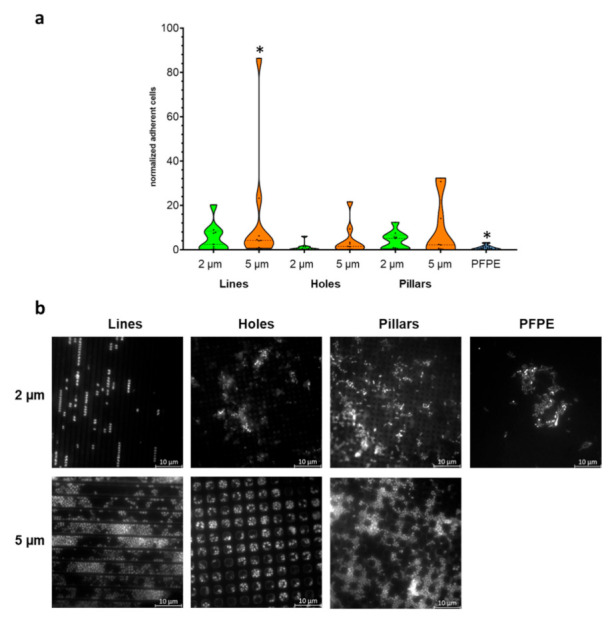
Adherence of microorganisms to micro-structured specimens. (**a**) Specimens made from PFPE containing microstructures of different geometries (lines, holes, or pillars) and dimensions (2 µm or 5 µm) as well as plain PFPE were exposed to the oral cavity for 8 h. Depicted is the number of adherent cells normalized to the adherence on plain PFPE. * *p* < 0.05 (**b**) Representative images of DAPI-stained specimens for the different structures and dimensions. Microorganisms are mostly detected within cavities.

**Figure 3 ijms-23-01157-f003:**
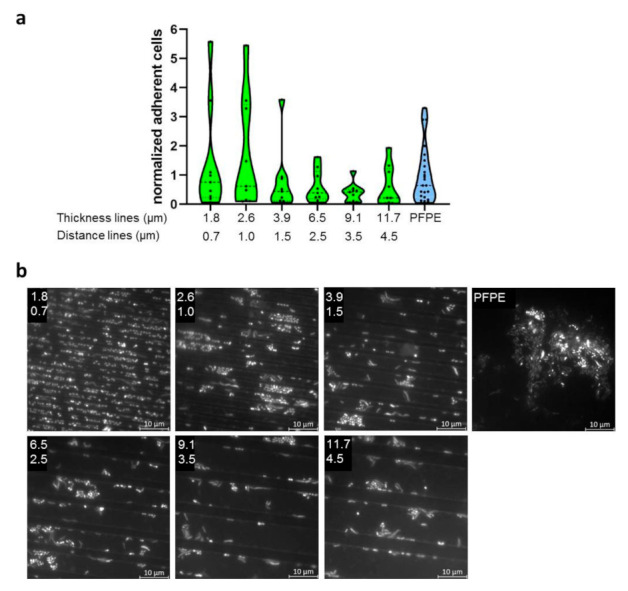
Adherence of microorganisms to specimens with line structures of different dimensions. (**a**) Specimens made from PFPE containing linear structures of different dimensions were exposed to the oral cavity for 8 h. Here, only selected dimensions are shown, with the full dataset available in Appendix A. Depicted is the number of adherent cells normalized to the adherence on plain PFPE. (**b**) Representative images of DAPI-stained specimens for the analyzed dimensions. The upper number indicates the thickness of the lines in µm and the lower number the distance of the lines in µm. Microorganisms are mostly detected within the recessed linear structures.

**Figure 4 ijms-23-01157-f004:**
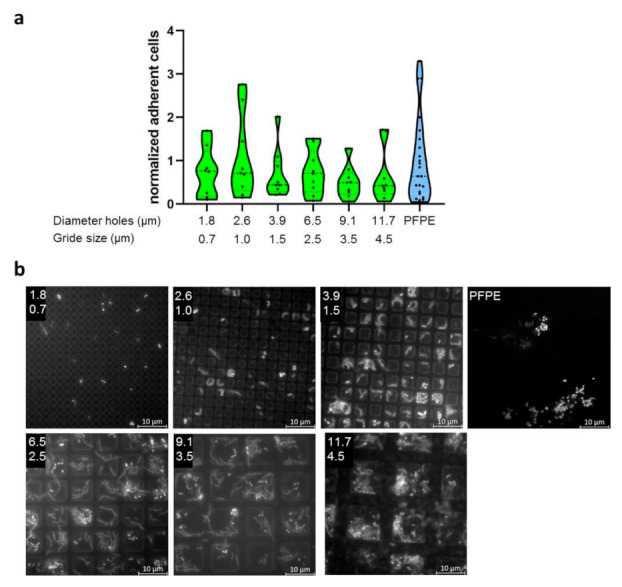
Adherence of microorganisms to specimens with hole structures of different dimensions. (**a**) Specimens made from PFPE containing holes of different dimensions were exposed to the oral cavity for 8 h. Here, only selected dimensions are shown, with the full dataset available in Appendix A. Depicted is the number of adherent cells normalized to the adherence on plain PFPE. (**b**) Representative images of DAPI-stained specimens for the analyzed dimensions. The upper number indicates the diameter of the holes in µm and the lower number the grid size in µm. Microorganisms are mostly detected within the recessed hole-like structures.

**Figure 5 ijms-23-01157-f005:**
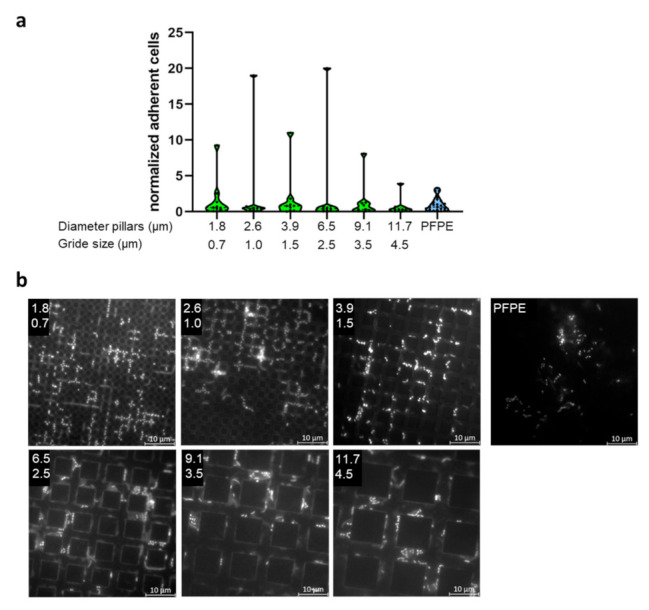
Adherence of microorganisms to specimens with pillar structures of different dimensions. (**a**) Specimens made from PFPE containing pillars of different dimensions were exposed to the oral cavity for 8 h. Here, only selected dimensions are shown, with the full dataset available in Appendix A. Depicted is the number of adherent cells normalized to the adherence on plain PFPE. (**b**) Representative images of DAPI-stained specimens for the analyzed dimensions. The upper number indicates the diameter of the pillars in µm and the lower number the grid size in µm. Microorganisms are mostly detected within the crevities between the pillar-like structures.

**Figure 6 ijms-23-01157-f006:**
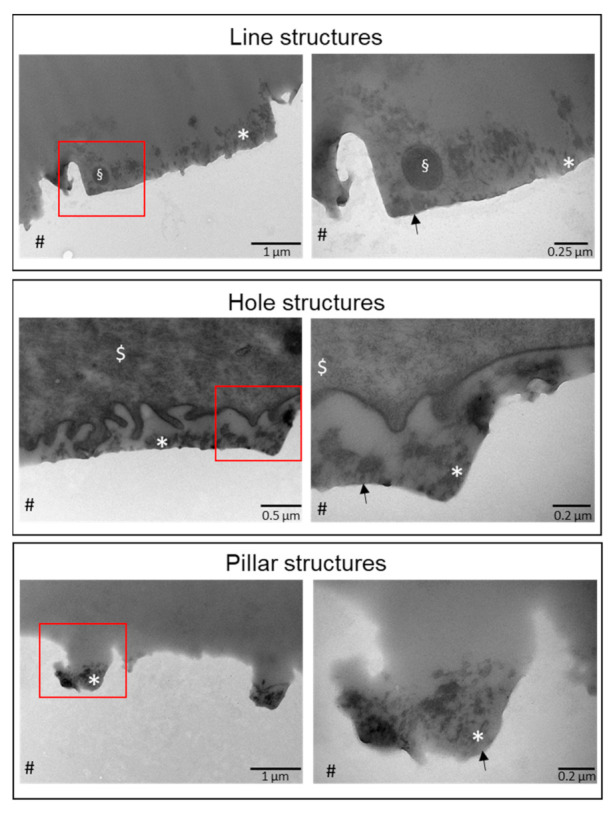
TEM images of specimens exposed to the oral cavity. Specimens made from PFPE containing lines, holes, or pillars were exposed to the oral cavity for 8 h. TEM images were prepared from the probes within the central area. Images on the left side show smaller magnifications while images on the right side depict larger magnifications of the area, indicated by a red square. # denotes the PFPE side. The whole surface was covered by a basal pellicle layer (arrow), however, globular and granular structures are, to the most part, only visible within the recessed areas (*). Additionally, bacteria (§) or epithelial cells ($) can be observed.

## Data Availability

The data presented in the current study are available on request from the corresponding author.

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
