# Peer review of "Bioadhesion on Textured Interfaces in the Human Oral Cavity—An In Situ Study"

_ijms, 2022, doi:10.3390/ijms23031157_

Round 1
Reviewer 1 Report
This study examines pellicle formation and bacterial colonization on patterned substrata. In contrast to many studies, the data were collected on specimens incubated in the oral cavity. The literature is in conflict as to the efficacy of substratum patterning on bacterial colonization, although several studies report trends. This aspect is covered nicely in the introduction, which is well written and very useful in the context of the present study. The results of the study show that no clear differences in colonization can be detected across many different patterns. In that sense, the study presents negative results. However, these results are extensive and important in the greater question of patterning as a biofilm inhibition approach. Despite my general belief that the manuscript is a valuable contribution, I have many reservations noted below. Some of these can likely be addressed simply by expanding or clarifying Materials/Methods. A more problematic situation is presented by my perspective on cell-counting as opposed to area coverage.
L26-27 – “in parts” should be “in part” (throughout the manuscript)
L73 – The pellicle is generally considered to be maximally a few hundred nanometers in thickness. It seems unlikely a structure of this size would “level out surface roughness” in the context of bacterial attachment. Please explain. Also, do the data in your study support the “leveling out” concept?
L75 – replace annihilated with eliminated or abrogated
L98-99 – I think this should read “A base structure……”, otherwise it is confusing what is being used/utilized.
L99 – Would PFPE be a useful material for oral applications such as lesion filler, crowns, prosthetics, etc.?
L331 – Was the time of day for splint wear consistent, i.e., in the morning vs in the afternoon? How many subjects participated and do the results reflect all participants? Are some subjects over-represented, meaning their data are included more than once because they wore stents more than once?
L334-335 – Do you have data that show enzyme activities do not change significantly during this transport time? In other words, if the subject had worn the stent until arrival at the clinic in the early morning, would the enzyme activities differ from those measured on stents transported in tissue?
L336-357 – Does “before and after” mean at time zero (immediately after immersion in the assay solution) and then ten min later? Or do you mean that the initial solution measured and then it was placed on the specimen? Importantly, briefly explain what exactly is being used as sample. The test surface itself? Material removed from the test surface? I think you are incubating the test surface in the assay solution, then removing the solution to measure fluorescence. Otherwise, wouldn’t the presence of the specimen influence the readings? Are specimens rinsed prior to analysis? In other words, are you potentially measuring salivary activities in addition to enzymatic activity within the pellicle? I don’t understand the desorbed measurements. It sounds as if the specimen was placed in the solution, incubated for 10 min, removed and then whatever value was measured was declared to be the desorbed value when in fact it must have included activity of the adsorbed fraction. I think these questions could all be addressed through substantial rewriting of this section. It could be augmented by more extensive description in the supplementary material, but I feel an informative and reasonably complete description could be included here. Please do not refer the reader to previous publications.
L358-365 – Was cell counting done “by eye” from the images in Supp Fig S1b? After overnight stent wear, much of the biofilm is likely to be more than one cell layer thick (e.g., the image in column 1, row 4; 2.5 x 2.5) and it would be extremely difficult to accurately determine cell numbers using either an air-dried or a hydrated sample because one cannot distinguish individual cells in biofilms of this thickness. For that particular image, it seems you have calculated a lower number (on average; data set shown in column seventeen of S1a) than for other images such as column 1, row 1 (0.7 x 0.7). Area coverage is clearly greater in the first image than in the second, yet you calculate a lower cell number. This is partially why confocal microscopy is employed in biofilm research, any time the biofilm is more than one or two layers thick. Use of the technique as currently described would allow one, at best, to simply determine area coverage, not cell number.
L98-112 – See comment above regarding the methods. I am not sure what exactly these numbers represent. However, because they do not differ across samples or between pellicle vs desorbed, it may not be important.
L112-114 – I understand your point about pellicle formation, but isn’t a comparison of pellicle on PFTE vs pellicle on enamel important? In other words, how does the pellicle on enamel look?
L122-133 – Move to M&M. Consider moving to supplementary material because this level of detail is not important to the microbiology or enzymology.
L147-150 – This is good approach and, at the same time, I got an answer to my question regarding L331. Please provide more detail on how many subjects and which subjects had repeated measurements. This could be done in the text of M&M or in supplementary material.
L162-170 – As with 122-133, I feel this material including Fig 4 should be moved to M&M. In the results section, it would be important to state that only every second feature was used for the values shown.
Fig 8 – Given the level of degree of coverage shown in Fig 7, I am wondering the basis for the statement that bacterial cells were “occasionally” observed. Further, do you have any light microscopy images in which you can point out an epithelial cell? Is it possible that nuclei within those cells would be counted as bacteria?
L243 – change manuscript to study
L263 – Have you presented data on the ultrastructure of the pellicle on enamel in this study?
Reviewer 2 Report
General comment: The authors covered an interesting topic! But the manuscript in general is not very clear in English. Starting with the title, the authors are not understandable in what they want to convey to readers. After reading, this manuscript does not appear as a "Communication", but as an "Article"; it is very long to be called a communication.
- The referee suggests to the authors to reformulate the title changing it in an appropriate one, for example “An in situ study…” .
- The “Introduction” has several terminologies not suitable and needs to be revised. For example, using “biotic and abiotic surfaces” instead of “shedding and non-shedding…”. In lines 43, 44 & 48, “of vital interest” scientifically it doesn't sound good; also, for “topographies” and “sheltering”… Here, have been used long sentences in several places and this creates confusion during the reading.
- Lines 92-95, the aim of this study is not clear...
- The results seem to be more like a methodology and discussion than results, just in a few lines you can find the results of the study. In line 98, delete the long name of PEPE.
- The word “in situ” is better to be written in italics.
- The legends of the figures, where it has been performed the statistics, it is necessary to mention the type of statistics done, not mentioning the methodology, but describing each point of the figures.
- The authors need to maintain the same unit also for the hours, in some places it is expressed as “hours” and in other “h”.
- The discussion needs to be revised and correcting English. It is often repeated “In the present study”.
- After the final discussion, is better to add a new section for the “Conclusions”, to emphasize the main goal of this work.
- How can this study can serve to the clinicians?! This is necessary to be added somewhere.
- In line 320 delete the long name of PEPE.
- For point 4.2, do the authors have any photos or scheme to better illustrate the procedure?
- In line 364, why the authors have used “Microbia”? Maybe they can replace it with an appropriate one.
- Point 4.6 is very poor as a statistical analysis; not clear.
Round 2
Reviewer 1 Report
The authors have responded appropriately to my comments. I feel only one minor change is important at this point. Somewhere in Sec 2.3, it must be emphasized that the biofilms studied here are only one cell layer in thickness.
Author Response
Thank you for your valuable input to improve our manuscript. According to your suggestion, we added a sentence “Also to most part, biofilms were only one cell layer in thickness” to sec. 2.3 (l. 168-169).
Reviewer 2 Report
The authors have corrected and improved their manuscript!
Author Response
Thank you for your valuable input to improve our manuscript.